# Interpretable Meta-Reinforcement Learning with Actor-Critic Method

## Abstract

Meta-reinforcement learning (meta-RL) algorithms have successfully trained agent systems to perform well on different tasks within only few updates. However, in gradient-based meta-RL algorithms, the Q-function at adaptation step is mainly estimated by the return of few trajectories, which can lead to high variance in Q-value and biased meta-gradient estimation, and the adaptation uses a large number of batched trajectories. To address these challenges, we propose a new meta-RL algorithm that can reduce the variance and bias of the meta-gradient estimation and perform few-shot task data sampling, which makes the meta-policy more interpretable. We reformulate the meta-RL objective, and introduce contextual Q-function as a meta-policy critic during task adaptation step and learn the Q-function under a soft actor-critic (SAC) framework. The experimental results on 2D navigation task and meta-RL benchmarks show that our approach can learn an more interpretable meta-policy to explore unknown environment and the performance are comparable to previous gradient-based algorithms.

## 1 Introduction

Reinforcement learning problems have been studied for a long time and there are many impressive works that achieved human-level control in real world tasks (Mnih et al., 2013; Silver et al., 2017; Vinyals et al., 2019; Schrittwieser et al., 2019). These agents are trained separately on each task and may require huge sampled data and millions of trails. However, in a many real world tasks, the cost of sampling data is not negligible, thus we cannot give agent a large number of trails in environment. In contrast, human can laverage past experiences and learn new tasks quickly in few trails, which is very efficient. Many tasks in fact share similar structures that can be extracted as prior knowledge, e.g., shooting games aims to eliminate enemies with weapons in different environments, which can help agent generalize quickly through different tasks. Meta-learn (Thrun & Pratt, 2012) reinforcement learning tasks can be a suitable chioce.

Meta-reinforcement learning (meta-RL) aims to learn a policy that can adapt to the unknown environment within few interactions with environment. Meta-policy can be seen as a policy that can derive new a policy maximizes the performance in the new environment. Gradient-based algorithms in meta-RL (Finn et al., 2017; Stadie et al., 2018; Rothfuss et al., 2018; Liu et al., 2019) showed that a meta-policy can be obtained by reinforcement learning a policy adapted by few reinforcement learning steps. The experiment results suggests that gradient-based methods can learn to sample and utilize sampled data in some extent. Nevertheless, the learning style and learned meta-policy are still far from human. Human learns a new task by interacting with the task sequentially and efficiently. With the obtaining of environment data, human gradually understanding where to sampling data and how to utilize the sampled data to adjust the policy, while gradient-based algorithms use parallel sampling neglecting the relations between data. Sampling independently is not data-efficient, usually needs a number of stochastic trajectories to do plicy adaptation. This causes the agent relying on the stochasticity to sample and only learns how to utilize data.

Inspired by the human behavior, we propose a K-shot meta-RL problem that constrains on the data amount accessed by agent, e.g., adapting policy within only two trails. Low resource environment simulates the real world tasks that have high costs on data obtaining, therefore, requires agent to learn a stable strategy to explore environment. To address the K-shot problem, we also propose a contextual gradient-based algorithm using actor-critic method. The adptation step uses a trail buffer

$D$ to store all the transitions in K-shot sampling and optimizes expected value for the states in $D$. The meta-learning step optimizes the expected return performed by adapted policy while learning the value functions and context encoder using soft actor-critic (Haarnoja et al., 2018) objectives. We learn the policy with reparameterized objective that derives an unbiased meta-gradient estimation and reduces the estimation variance for Q-value. Our contribution can be summarized as follows:

- We reformulate and propose the K-shot meta-RL problem to simulate the real world environment.

- We propose a new gradient-based objective to address the K-shot problem.

- We introduce context based policy and value functions to perform efficient data sampling.

- We use actor-critic method to reduce the variance and bias of estimation in Q-value and meta-gradien.

## 2 RELATED WORK

Meta-reinforce learning algorithms mainly have three different categories: gradient-based motheds (Finn et al., 2017; Stadie et al., 2018; Rothfuss et al., 2018; Liu et al., 2019; Nichol et al., 2018), recurrent meta-learners (Wang et al., 2016; Duan et al., 2016), multi-task learners (Fakoor et al., 2019; Rakelly et al., 2019). Gradient-based algorithms like MAML (Finn et al., 2017) optimizing the policy updated by one step reinforcement learning, aiming at learning a good initialization of the policy weights. E-MAML (Stadie et al., 2018) considered the impact that the data obtained by meta-policy can influence the adapted policy's performance and assigned credit for meta-policy. While ProMP (Rothfuss et al., 2018) modified the adaptation gradient estimator to be low variance on second-order gradient. Recurrent meta-learners (Wang et al., 2016; Duan et al., 2016) use RNN as a meta-learner that can learn new task from environment data while exploring. The RNN learners are optimized with sequentially performed episodes end-to-end, which is more similar to the learning process of human and more interpretable in meta-policy. Multi-task learners (Fakoor et al., 2019; Rakelly et al., 2019) aim at learning multi-task objective to solve meta-learning problems. They argue that meta-learning can be done by explicitly resuing the learned features through context variable. MQL (Fakoor et al., 2019) can even perform well without adaptation. PEARL (Rakelly et al., 2019) constructs context encoder to infer the latent task variable and also learns a multi-task objective. The trained policy can perform structured exploration by inferring the task while interacting with environment.Our approach is related closely to the gradient-based researches which also tries to reduce the estimation variance and bias of the second-order gradient, however, we estimate the second-order gardient with value functions, and we still want perform structured exploration in data expensive environments.

## 3 BACKGROUND

This section focuses on the problem definition and notation of reinforcement learning and meta-reinforcement learning problems.

### 3.1 REINFORCEMENT LEARNING

Reinforcement learning (RL) problems aim to maximize the expectation of episode returns

$$\mathbb{E}_{\tau \sim P(\tau|\theta)}[R(\tau)] = \mathbb{E}_{\tau \sim P(\tau|\theta)}[\sum_t \gamma^t r(s_t, a_t)] \tag{1}$$

with single task and agent, where $\tau = \{s_0, a_0, r_0, \dots\}$ is the trajectory performed by the agent, $s_0 \sim \rho^0$ is the initial state, $a_t \sim \pi_\theta(a_t|s_t)$ is the action sampled from the policy $\pi$ that parameterized by $\theta$, $s_{t+1} \sim P(s_{t+1}|a_t, s_t)$ is the state at timestep $t$, and $P(s_{t+1}|a_t, s_t)$ is the transition probability. The problem can be represented by a Markov Desicion Process (MDP) with tuple $\mathcal{M} = (\mathcal{S}, \mathcal{A}, \mathcal{P}, \mathcal{R}, \rho^0, \gamma, H)$, where $\mathcal{S} \subseteq \mathbb{R}^n$ is the set of states, $\mathcal{A} \subseteq \mathbb{R}^m$ is the set of actions, $\mathcal{P}(s'|s, a) \in \mathbb{R}^+$ is the system transition probability, $\mathcal{R}(s, a) \in \mathbb{R}$ is the reward function of the task, and $H$ is the horizon.

Optimizing (1) usually uses gradient descent and the gradient is estimated using vanilla policy gradient (VPG) estimator (Williams, 1992)

$$\nabla_\theta \mathbb{E}_{\tau \sim P(\tau|\theta)}[R(\tau)] = \mathbb{E}_{\tau \sim P(\tau|\theta)}[\nabla_\theta \log \pi(\tau) R(\tau)]$$

$$\approx \frac{1}{N} \sum_i \sum_t \nabla_\theta \log \pi_\theta(a_{i,t}|s_{i,t})(\sum_{t'=t}^{H} R(s_{i,t'}, a_{i,t'})) \tag{2}$$

## 3.2 GRADIENT-BASED META-REINFORCEMENT LEARNING

Meta-reinforcement learning (meta-RL) aims to learn a fast adaptation procedure that can leverage the learned prior knowledge from training tasks and adapt to new tasks with few steps. A task $T$ in meta-RL can also be defined by an MDP $\mathcal{M}_T = (\mathcal{S}, \mathcal{A}, \mathcal{P}_T, \mathcal{R}_T, \rho^0, \gamma, H)$. The task is drawn from a distribution $T \sim P(T)$, for simplicity, we only consider tasks with different reward functions or system transitions but the same state and action space.

Gradient-based meta-RL algorithms (Finn et al., 2017; Stadie et al., 2018) are mainly based on the basic meta-objective (Rothfuss et al., 2018)

$$J(\theta) = \mathbb{E}_{T \sim P(T)}[\mathbb{E}_{\tau' \sim P_T(\tau'|\theta')}[R(\tau')]], \quad \theta' = U(\theta, T) = \theta + \alpha \nabla_\theta \mathbb{E}_{\tau \sim P_T(\tau|\theta)}[R(\tau)], \tag{3}$$

where $\theta$ is the weights of meta-policy, and $\theta'$ is the adapted weights after one step gradient descent. The meta-objective $J(\theta)$ optimizes the expectation of episode return sampled from the adapted policy $\pi_{\theta'}$. The meta-gradient can be written as

$$\nabla_\theta J(\theta) = \mathbb{E}_{T \sim P(T)}[\mathbb{E}_{\tau' \sim P_T(\tau'|\theta')}[\nabla_{\theta'} \log P_T(\tau'|\theta') R(\tau') \nabla_\theta \theta']]$$

$$\nabla_\theta \theta' = I + \alpha \nabla_\theta^2 \mathbb{E}_{\tau \sim P_T(\tau|\theta)}[R(\tau)] \tag{4}$$

# 4 METHOD

## 4.1 REFORMULATE META-REINFORCEMENT LEARNING PROBLEM

Different tasks have different features in MDP, a task can be inferred from few important states and transitions in the environment, e.g., different friction coefficients on floor, different rewards for the same state and action, or some states only exists in certain environments. We name these states and transitions the *feature points* of the environment. Humans usually learn a task sequentially and efficiently since they can easily recognize the feature points in an environment. The exploration policy of a human changes significantly after obtaining data from the envioronment, thus they can decide where to explore and learn a task quickly. However, as formula (3), fast adaptation $U(\theta, T)$ usually refers to few gradient descent steps in initial weights $\theta$, and unlike humans, the updating is performed in a batched style as normal reinforcement learning. Batched sampling usually contains a large number of trajectories in parallel, which can be inefficient for inferring the task. E-MAML (Stadie et al., 2018) also tried to improve the sampling efficiency of meta-policy by accounting for the fact that samples drawn from meta-policy will impact the adapted policy. Inspired by the learning procedure of human, we reformulate the meta-RL problem as *K-shot meta-reinforcement learning*.

**Definition.** Given a task $T \sim P(T)$, the agent samples data in trail phase and perform good policy in test phase. In trail phase, the agent can only sequentially sample K trajectories in total to adjust its policy, with each trajectory of $H$ length. In test phase, the agent is required to perform only one trajectory and make the return as high as possible.

K-shot meta-RL problem defined above constrains the amount of data that can be accessed by agent, and is more similar to the real world meta-RL problem, e.g., super mario maker. In K-shot setting, meta-policy can still be updated using $U(\theta, T)$ with batched trajectories, since they can be seen as sampled independently in sequence. However, the variance of the gradient estimation grows as K descends, which means the performance becomes more unstable. To optimize the problem, we propose a new meta-objective

$$J^{K-shot}(\theta) = \mathbb{E}_{T \sim P(T)}[\mathbb{E}_{\tau' \sim P_T(\tau'|\theta')}[R(\tau')]],$$

$$\theta' = U(\theta, D)$$

$$= \theta + \alpha \nabla_\theta \mathbb{E}_{s \sim D}[V^\pi(s|c)] \tag{5}$$

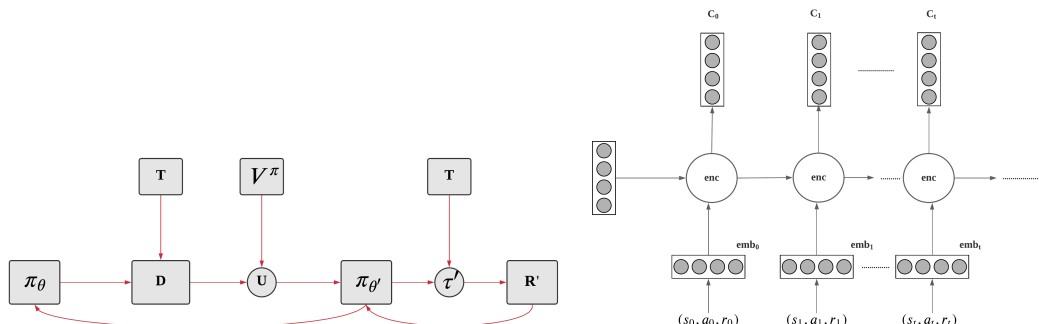

Figure 1: **Left**: The whole computation graph for K-shot meta-RL, where $\theta$ is the meta-policy parameter, $D$ is the trail buffer storing transitions and context in trail phase, $T$ is the task sample from task distribution $P(T)$, $V^\pi$ is the value function evaluating policy $\pi$. Straight lines represent for forward path and curved lines are backward path. K-shot Meta-RL optimizes the average return after policy adaptation using trail buffer. **Right**: The LSTM context encoder structure. A transition is first embeded by a linear layer and fed through LSTM to form context $c_t$ at timestep $t$. The hidden state will be kept between different episodes.

for the K-shot setting. Here $D$ is the state buffer sampled by meta-policy in trail phase, and $V^\pi(s|c)$ is the expected return of policy $\pi$ at state $s$ under context $c$ (see 4.2 for details). The state buffer $D$ contains $K * H$ states as described in definition, which means the agent can only use few states to update its policy. Due to the constraint on availble environment information, the agent is encouraged to learn to explore more important states that can help performing well in test phase.

## 4.2 INTRODUCING CONTEXT

In meta-RL, the task $T$ sampled from the task distribution is not given to the agent and can be thought of a latent variable of the task MDP. The latent variable can be inferred and has a strong correlation with the context variable $c$ which is encoded by the experience $(s_t, a_t, r_t, s_{t+1})$ collected until time step $t$. The context variable contains the information of two aspects. First, the experience is one step of transition and reward that represents the system dynamics and reward function of the environment. Second, the decision history $\{(s_t, a_t)\}_n$ represents the agent policy in the current environment. Q-function uses the state action tuple $(s_t, a_t)$ to evaluate the future discounted return of the policy at state $s_t$ taking action $a_t$, which also need the same two-aspect information about policy and dynamics. Therefore, we introduce a contextual Q-function $Q(s, a|c)$ that can evaluate policy in unknown environment. To encourage the agent to learn how to sample efficiently and infer the task from the unknown environment, the agent should also use a context depended policy $\pi_\theta(a|s, c)$ to memorize past states.

Encoding the context variable $c$ uses a Long Short Term Memory (LSTM) (Hochreiter & Schmidhuber, 1997). The context encoder takes as input the history experience so far and output a context variable $c$ deterministicly. LSTM encoder has an advantage of dealing with sequential data like history transitions, thus can give a good representation of context. Addationally, LSTM context can represent for the same current state with different history states, which helps agent to explore more states and Q-function to evaluate the discounted return correctly.

We follow the setting in (Duan et al., 2016) to design the context encoding. Transitions are continuously fed into LSTM encoder while agent performing trajectories. The initial context is a zero vector and the context will not be reset after episode ends. This means agent can keep the information between episodes and decide how to explore in next steps. With setting, the adaptation procedure is divided into two parts. First, the agent samples important states for itself in environment according to the data collected so far. Second, the agent uses all data available in buffer $D$ to adapt policy. Through this process, agent can learn how to explore environment and how to utilize the transition data, which is a more structured learning scheme.

### 4.3 LEARNING WITH ACTOR-CRITIC METHOD

Solving the K-shot problem in 4.1 requires value functions to evaluate the future expected return of policy $\pi$, therefore, training the agent in an actor-critic style can be a good choice. The adaptation step in (3) uses reward term to estimate the Q-value. Even it is an unbiased point estimation of $Q(s_t, a_t)$, the variance have can be very high (Konda & Tsitsiklis, 2000), and may lead to an unstable learning process. Actor-critic algorithms can trade-off between variance and bias of the estimation and the learned value functions can be used to do adaptation.

To learn the value functions, we use soft actor-critic (SAC) (Haarnoja et al., 2018) framework. SAC is an off-policy RL algorithm that tries to learn a policy with maximized entropy, thus the agent can trad-off between exploration and exploitation. We modified the SAC objective as

$$
\begin{aligned}
J^{SAC}(\theta) &= \mathbb{E}_{s \sim D}[V^\pi(s|c)] \\
&= \mathbb{E}_{s \sim D, a \sim \pi_\theta(a|s,c)}[Q^\pi(s,a|c) - \alpha \log \pi_\theta(a|s,c)] \\
&= -\mathbb{E}_{s \sim D}[D_{KL}(\pi_\theta(\cdot|s,c) \,\|\, \exp(\frac{Q^\pi(s,\cdot|c)}{\alpha}))],
\end{aligned}
\tag{6}
$$

adding context dependency to value functions and policy, and the value functions also satisfies Bellman equation

$$
Q^\pi(s_t, a_t|c_t) = R(s_t, a_t) + \mathbb{E}_{s_{t+1} \sim P(s_{t+1}|s_t, a_t)}[V^\pi(s_{t+1}|c_{t+1})]
\tag{7}
$$

and

$$
V^\pi(s_t|c_t) = \mathbb{E}_{a_t \sim \pi_\theta(a_t|s_t, c_t)}[Q^\pi(s_t, a_t|c_t) - \alpha \log \pi_\theta(a_t|s_t, c_t)]
\tag{8}
$$

where $c_{t+1} = enc(s_t, a_t, r_t|c_t)$ and $enc$ is the LSTM context encoder mentioned in 4.2.

Learning Q-function, V-function and LSTM encoder requires minimizing loss

$$
L^Q = \mathbb{E}_{(s,a,s') \sim D}[(Q^\pi(s,a|enc(\tau_{1:t-1})) - (r(s,a) + \gamma \hat{V}^\pi(s'|enc(\tau_{1:t}))))^2]
\tag{9}
$$

and

$$
L^V = \mathbb{E}_{s \sim D}[(V^\pi(s|c) - \mathbb{E}_{a \sim \pi_\theta(a|s,c)}[Q(s,a|c) - \alpha \log \pi_\theta(a|s,c)])^2]
\tag{10}
$$

where $D$ is the replay buffer (Mnih et al., 2015) storing the transitions experienced by agent, $s$ is the state, $a$ is the action taken at state $s$, $s'$ is the next state given state and action $(s,a)$, $r(s,a)$ is the reward at state $s$ after taking action $a$, $\tau_{1:t-1}$ and $\tau_{1:t}$ represents the trajectory before state s and including state s, and $\hat{V}$ is the target value function to stable value iteration.

Substitue the adaptation objective in (5) with (6), we have

$$
U(\theta, D) = \theta + \alpha \nabla_\theta \mathbb{E}_{s \sim D, a \sim \pi_\theta(a|s,c)}[Q^\pi(s,a|c) - \alpha \log \pi_\theta(a|s,c)],
\tag{11}
$$

where $c$ refers to the context at state $s$. The gradient estimation in second term using VPG estimator is

$$
\begin{aligned}
&\nabla_\theta \mathbb{E}_{s \sim D, a \sim \pi_\theta(a|s,c)}[Q^\pi(s,a|c) - \alpha \log \pi_\theta(a|s,c)] \\
&\approx \frac{1}{N} \sum_i \nabla_\theta \log \pi_\theta(a|s,c)(Q^\pi(s,a|c) - \alpha \log \pi_\theta(a|s,c))_\perp
\end{aligned}
\tag{12}
$$

Here $\perp$ means stop gradient. However, the second-order gradient of the analytical form and the Monte Carlo approximation form are not the same, which are

$$
\mathbb{E}_{s \sim D, a \sim \pi_\theta(a|s,c)}[(\nabla_\theta^2 \log \pi_\theta(a|s,c) + \nabla_\theta \log \pi_\theta(a|s,c)^2)(Q^\pi(s,a|c) - \alpha \log \pi_\theta(a|s,c))]
\tag{13}
$$

and

$$
\frac{1}{N} \sum_i \nabla_\theta^2 \log \pi_\theta(a|s,c)(Q^\pi(s,a|c) - \alpha \log \pi_\theta(a|s,c))_\perp.
\tag{14}
$$

This will cause a biased estimation in meta-gradient. Suppose policy $\pi_\theta$ is a Gaussian distribution, the action can be rewritten as a deterministic form $a = \mu_\theta(\epsilon; s|c)$, where $\epsilon \sim N(0; 1)$, and the gradient term in (11) can be reparameterized as

$$
\nabla_\theta \mathbb{E}_{s \sim D, \epsilon \sim N(0;1)}[Q^\pi(s, \mu_\theta(\epsilon; s|c)|c) - \alpha \log \pi_\theta(\mu_\theta(\epsilon; s|c)|s, c)]
\tag{15}
$$

---

**Algorithm 1** K-shot Meta-Reinforcement Learning

---

**Require:** trials $K$, horizon $H$, task distribution $P(T)$, learning rates $\alpha$, $\beta$, $\delta$

    Initialize trail buffer $\hat{D}^i$ and replay buffer $D^i$ for each training task

    Initialize weights of $\mu_\theta$, $enc_\phi$, $Q_\psi$, $V_\eta$, $\hat{V}_{\eta'}$

1: **while** not done **do**
2:   **for** i=1,2,...,N **do**
3:     Clear trial buffer $\hat{D}^i$
4:     Sample $T_i$ from $P(T)$
5:     Sample $K$ trajectories from $T_i$ with $\mu_\theta$ while encoding experiences,
       and add to $\hat{D}^i$ and $D^i$
6:     Compute adapted policy using $\theta' = U(\theta, \hat{D}^i)$ in (17)
7:     Run adapted policy $\theta'$ for several turns to estimate average return $R(\tau')$,
8:     Compute meta-gradient $\nabla_\theta J_i^{K-shot}(\theta)$
9:     Sample a batch of transitions in $D^i$
10:     Compute gradients $\nabla_\phi L_i^Q$, $\nabla_\psi L_i^Q$, $\nabla_\eta L_i^V$ using sampled batch
11:   **end for**
12:   $\theta \leftarrow \theta + \beta \frac{1}{N} \sum_i \nabla_\theta J_i^{K-shot}(\theta)$
13:   $\psi \leftarrow \psi + \beta \frac{1}{N} \sum_i \nabla_\psi L_i^Q$
14:   $\eta \leftarrow \eta + \beta \frac{1}{N} \sum_i \nabla_\eta L_i^V$
15:   $\phi \leftarrow \phi + \beta \frac{1}{N} \sum_i \nabla_\phi L_i^Q$
16:   Soft update target $\eta' \leftarrow (1 - \delta)\eta' + \delta\eta$
17: **end while**

---

Thus the second-order gradient of Monte Carlo approximation

$$\frac{1}{N}\sum_i (\nabla_a^2 Q(s,a|c)\nabla_\theta \mu_\theta(\epsilon_i; s|c)^2 + \nabla_a Q(s,a|c)\nabla_\theta^2 \mu_\theta(\epsilon_i; s|c)$$
$$-\alpha(\nabla_a^2 \log \pi_\theta(a|s,c)\nabla_\theta \mu_\theta(\epsilon_i; s|c)^2 + \nabla_a \log \pi_\theta(a|s,c)\nabla_\theta^2 \mu_\theta(\epsilon_i; s|c))) \tag{16}$$

is an unbiased estimation of the analytical form, and from (4) we know that meta-gradient estimation can be unbiased using this adaption form. To utilize all the available data in $D$, we use deterministic form for adaptation step in (5) and rewrite K-shot meta-objective as

$$J^{K-shot}(\theta) = \mathbb{E}_{T\sim P(T)}[\mathbb{E}_{\tau'\sim P_T(\tau'|\theta')}[R(\tau')]],$$
$$\theta' = U(\theta, D)$$
$$= \theta + \alpha \nabla_\theta \frac{1}{N} \sum_{i=1}^{|D|} (Q^\pi(s_i, \mu_\theta(\epsilon_i; s_i|c_i)|c_i) - \alpha \log \pi_\theta(\mu_\theta(\epsilon_i; s_i|c_i)|s_i, c_i)) \tag{17}$$

where $s_i$, $\epsilon_i$, $c_i$ are the $i^{th}$ data in replay buffer collected at trail phase. The meta-RL problem proposed in 4.1 can be directly optimized by (17) while learning value functions with (9) and (10).

## 5 EXPERIMENTS

To evaluate our algorithm proposed above, we implemented our approach in several different meta-reinforcement learning environments, including 2d navigation task from (Rothfuss et al., 2018) and meta-RL benchmarks previously used by (Rakelly et al., 2019) in mujoco (Todorov et al., 2012).

### 5.1 ENVIRONMENT SETUP

First we introduce the 2d navigation task. This task requires the agent to explore a sparse reward environment, infer the goal point in an unbounded 2d plane. The plane is divided into four parts with each goal in one part. The agent starts from center of the plane and tries to obtain task data in trail phase and reach the goal in test phase. The observation to be received is its coordinate concatenating the remained available steps. The reward is sparse and set to be the difference between the distances

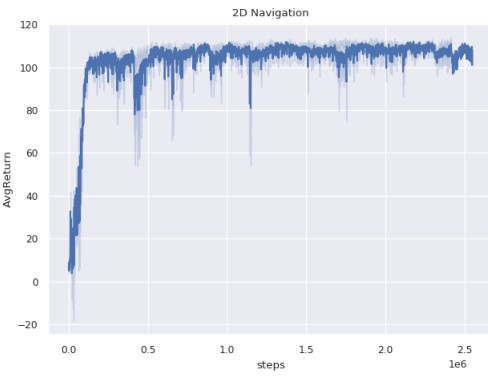

Figure 2: The 2d navigation task result. The policy converged, and the performance is close to max score in task.

to the goal within two steps when near the goal, otherwise is set to zero. We use this environment to test whether the agent can learn to sample different states and use these states to adapt to the right policy. Second, we describe the mujoco benchmarks. Mujoco tasks are environments for controlling robot in simulated physical world to learn task adaptation. We tested three mujoco environments: HalfCheetahForwardBack, AntRandDir, HalfCheetahRandVel. HalfCheetahForwardBack requires the agent to run forward or backward as fast as possible, AntRandDir requires agent to run in two random selected directions as fast as possible, and HalfCheetahRandVel requires agent to run with certain speeds.

## 5.2 RESULTS

In this section we will show the experiment results of our approach. In 2d navigation task, the meta-policy learning curve converged easily in early training steps and the trained meta-policy is shown in figure 3. The agent have three trails on each task, then perform a apdated policy for testing. In each trail, the agent perform 100 steps and total 300 steps in trail phase. This means our approach uses less steps to figure out the task than it is in Rothfuss et al. (2018) which used 2000 steps. As is shown in figure 3, agent chose very different states to explore. Each trail the agent will visit the states that have not been explored, and states are separated in plane with clear bounds. These states helps the agent to infer the task efficiently, and the meta-policy can be performed in data expensive environments.

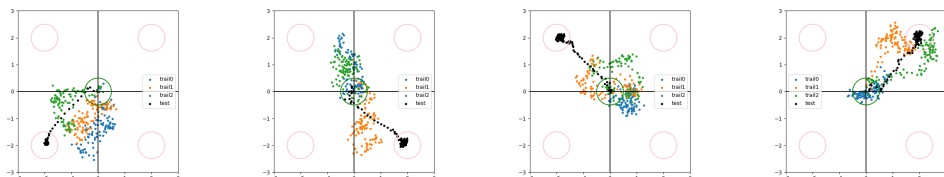

Figure 3: meta-policy and adaptation in 2d navigation, where pink circles are the goals green circle is zero reward area. Agent not in the right part corresponding the current goal or in green circle will get zero reward. In each task agent has three trails to collect data, then perform the test phase.

We also evaluated our algorithm in mujoco meta-RL environments. The results[1] are showed in figure 4. The performances of our algorithm are slightly higher than the previous gradient-based algorithms. The data amount we sampled at trail phase are also less than it is in MAML, ProMP and even in PEARL (Rakelly et al., 2019). Each trail we sampled 200 steps for total 2 trails, the data used to do adaptation in our algorithm is 10% of in PEARL and 5% in ProMP.

---

[1]The MAML and ProMP results are obtained from published results in Rakelly et al. (2019)

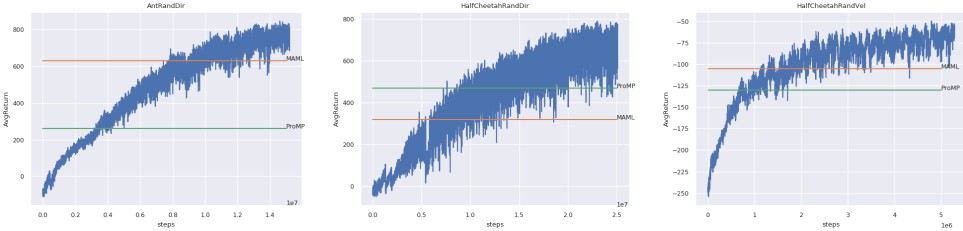

Figure 4: meta-policy and adaptation in mujoco environments. The performance is compared with previous gradient-based algorithms. The performance is better than the previous algorithms.

## 6  CONCLUSION

In this paper, we proposed a new meta-RL problem that contrains the data amount utilized by agent, and have given a new meta-RL algorithm that optimizing with contextual policy and actor-critic framework. Our approach can estimate unbiased meta-gradient and reduce the estimation variance of Q-function. From the experiments, we demonstrated that contextual policy can sample efficiently in data constrained environments. Finally, the experiments on mujoco environments suggested that our algorithm can have competitive performance with other gradient-based algorithms. Human behavior can usually bring us inspiration on designing the intelligent system and maybe is a key to AGI.

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
