# OpenReview forum: "Interpretable Meta-Reinforcement Learning with Actor-Critic Method"
_ICLR.cc/2021/Conference — Reject_

### Official Review · AnonReviewer3 · 2020-10-18
**Promising idea, but the work is in an unfinished state right now.**

**Rating:** 4
**Confidence:** 3

**Review:**

### **Summary and Contributions of Paper**
This paper proposes to improve the K-shot RL meta-learning problem by using an LSTM, whose repeated inputs are (s,a,s') state-action transitions, and whose step-wise outputs are context vectors. The policy, during the K-shot phase, then additionally observes the current context vector along with the standard state, in order to produce an action.

In order to train both the policy and the LSTM in an end-to-end fashion, the paper modifies the MAML objective appropriately, since the LSTM's context vector is used throughout the entire K-shot process.

Experiments are performed on the Nav-2D 4-corner exploration task (proposed by ProMP, 2018) and a few standard Mujoco meta-learning tasks.

### **Strengths**
- Seems to be novel in the sense of allowing an RNN to control how the policy explores the task during the K-shot phase. This is in contrast with previous works which used manual methods for the policy exploration.
- Once I understood Figure 3 (past the presentation issues), the LSTM's context vector seems to signal to the policy which areas to explore next (rather than entropy-dependent exploratory movement in the default MAML algorithm) in the K-shot phase, which is good exploration behavior.


### **Weaknesses**
- One of the most obvious issues are the numerous grammar issues/awkward phrasings. The writing of the paper can be significantly improved. This also includes the mathematical notation, as variables are sometimes used abruptly without introduction. I do think this paper has promise, but really needs to be cleaned up in its presentation.
- Experiments can be more comprehensive. So far, there are only 3 Mujoco tasks used, whereas there are 8+ to choose from (see T-MAML, 2019). More Mujoco benchmarks will make the results convincing. Furthermore, the authors only plotted the recorded scores from previous papers (as horizontal lines) rather than run the baselines themselves. One issue here is that in many cases, there are multiple task definitions/tweakings that are not the same across papers. For instance, the original MAML paper and T-MAML used completely different scalings for the alive bonus, movement reward, and the energy expenditure penalties for the task "ForwardBackwardAnt." The authors need to run the baselines themselves in order to have rigorous metrics. This should not be difficult, as there are already very clean implementations, e.g. see (https://github.com/lhao499/taming-maml).

I think the paper proposes some promising ideas. However, due to the weaknesses mentioned, I think the paper needs to be cleaned up a lot more before being ready for submission, and thus I propose rejection.


### **Clarity Questions**
- Can you provide some explanations for pros/cons against PEARL (which also uses a context vector approach, but uses a feed-forward VAE-like network for the context vector?) PEARL seems to be the most direct competitor to the proposed paper's approach, as both are using context vectors.

---

### Official Review · AnonReviewer2 · 2020-10-26
**The paper is hard to understand and contains unjustified claims**

**Rating:** 3
**Confidence:** 4

**Review:**

Authors introduce a new meta-RL algorithm based on SAC. It uses a context variable $c$ that they condition the Q-function on and the adaptation mechanism which is based on the values of the value function (ie. $\mathbb{E_a} Q(\dot, a)$) instead of the true returns. Authors claim their method reduces variance and bias of the meta-gradient estimation, is closer to human learning, encourages the agent to learn to explore, is more data-efficient in test-time and has competitive performance among gradient-based algorithms.

I found many of the claims of the paper are unjustified:
1. Authors claim "we reformulate and propose the K-shot meta-RL problem to simulate the real world environment". The formulation that follows is the standard one (cf. MAML (Finn et al. 2017): "In K-shot reinforcement learning, K rollouts from $f_\theta$ and task $T_i$ (...) may be used for adaptation on a new task $T_i$". It is unclear how does the authors' definition relate to the real world.
2. They suggest their method is able to "learn where to explore", but nothing in the method specifically addresses this part (compared to eg. Learning to reinforcement learn (Wang et al. 2016), which also has a learned context that influences the policy).
3. They suggest their method learns more "human-like", which is understood as "not using batched sampling" (as, arguably, humans learn more sequentially). However, their method also uses batched sampling (see eq. (17)). In another part of the paper human-like learning is associated with the usage of LSTM, which is also neither novel (see Wang et al. (2016) again) nor grounded.
4. They claim their method decreases bias and variance. Gradient estimation in a typical meta-RL method is unbiased, so that's hard to decrease. On the other hand, I believe the variance of the proposed method is increased compared to E-MAML and others due to approximating rewards with V, which, due to the fixed-capacity of the model, will make their adaptation procedure inherently biased and thus estimating gradients in incorrect places.
5. Their method "makes the meta-policy more interpretable", yet no interpretation attempt was presented.
6. "Agent can learn how to explore environment and how to utilize the transition data, which is a more structured learning scheme". The learning scheme is basically the same as in PEARL (Rakelly et al. 2019). It's not clear what exactly makes the learning scheme "more structured" nor what it means.
7.  "Each trail we sampled 200 steps for total 2 trails, the data used to do adaptation in our algorithm is 10% of in PEARL and 5% in ProMP.". According to Rothfuss et al. (2018), app. D.2, they used a single trajectory of 200 steps for adaptation for HalfCheetahFwdBack, which is half of what the authors' method used. On the other hand, PEARL achieves much better results than the authors' method, what is not mentioned in the paper.

Details of the method are not very clear, I assume they follow SAC with online/target networks, but am not really sure nor what $\mu$, $\phi$ and $\eta$ mean. Mechanics of meta-testing also weren't described in enough detail.

The writing of the paper is terrible, to the point it's not always clear what the authors mean. Most of the problems are simple grammar errors, which are easily solvable by a tool like grammarly or google docs correction. As such, I consider sending the paper to the review in its current state a disrespect for the reviewers who have to spend the time to decipher the writeup.

Grammar problems (first page only):
1. However, in many real world tasks
2. Derive a new policy **which** maximizes
3. The experiment results suggest that
4. Utilize sampled data **to** some extent.
5. **While** obtaining environment data
6. Human gradually understand**s** where to sample data
7. Stochastic trajectories to do p**o**licy adaptation
8. "This causes agent (...) and only learns how to utilize data" - not clear what was meant
9. That have high cost o**f** obtaining data

---

### Official Review · AnonReviewer4 · 2020-10-29
**Conceptual and presentation issues throughout the paper**

**Rating:** 4
**Confidence:** 4

**Review:**

The paper presents a new meta reinforcement learning algorithm that uses trajectory information to create a contextual embedding for each task. The context is used as a given condition for learning the Q function and policy using soft actor critic algorithm.

Suggestions to improve the quality of the paper:
- The title states "interpretable" but there is no interpretability discussed anywhere in the paper. A more relevant title would be appropriate.
- The idea of contextual embedding using trajectory and LSTMs have been explored in multiple papers (e.g. PEARL (Rarely et al. 2019), MQL (Fakoor et al., 2019). It is not clear how the proposed method is different from these.
- It is unclear how adding context information to the Q function leads to structured exploration. SAC does not use Q function to explore, it uses the probability distribution of the policy network output for exploration. Prior work has shown that SAC exploration can be improved with dual Q function and upper confidence bounds, but that is not being used here: https://papers.nips.cc/paper/8455-better-exploration-with-optimistic-actor-critic
- It is unclear how the K-shot version of the meta-RL algorithm is different from other meta-RL algorithm. The proposed algorithm also samples from the tasks uniformly, same as prior algorithms.
- There are numerous typographical and grammatical errors throughout the paper. These can be easily fixed with modern text editors. Some examples - trails, data in some extent, where to sampling data, plicy, meta-gradien, want perform, availble, help performing, subistitue, contrains, etc.
- Figure 3 and 4 are very difficult to read, even on a computer. The graph title, axes labels, axes tick marks, legend are all ineligible. The quality of these graphs need to be improved for readability.
- The evaluation results are over two trials, need more of them for a convincing argument that the proposed method is better than prior methods.

---

### Official Review · AnonReviewer1 · 2020-10-30
**My review**

**Rating:** 2
**Confidence:** 5

**Review:**

Summary:

This paper proposes a new approach for meta-RL. The paper claims that the proposed method reduces variance and bias of the meta-gradient estimation by only a few samples. In addition, this paper claims that their method is more interpretable.

My comments:

There are lots of unknown, unwarranted claims about this paper in addition to no thorough experiments and comparison with previous works :

1. Paper claimed that their proposed method not only reduces the variance of meta-gradient but also reduces bias. Reading through this method and experiments, I don't see any theoretical or empirical justification why that should be the case.

2. The method claims this method has a more interpretable meta-gradient. Again, there is nothing in this paper that verifies this claim.
3. Paper proposed to use fewer samples for the adaptation phase. I don't understand why this can help at all.
4. Experiments are incomplete and there is no rigorous evaluation to analyze this method.
5. The proposed context in this paper has been already proposed in previous work, not sure what is new here.
6. There are lots of things that need to defined like trail, few-shot, etc.

In summary, this paper is not ready at all and needs lots of works. Hence, I'd recommend this paper to be rejected.

---

### Official Review · AnonReviewer5 · 2020-11-07
**Review 5**

**Rating:** 3
**Confidence:** 3

**Review:**

Summary
----------

This paper presents an approach to meta-RL based on combining gradient-based updating with recurrence-based meta-learning. The approach is based on combining gradient-based policy updating with recurrence-based updating of the value function. The authors evaluate the method of simple standard meta-RL benchmarks.


Comments
----------

Overall, the direction of the paper is interesting but the paper has numerous shortcomings.

First, the proposed method is a fairly straightforward combination of existing techniques. In particular, the approach consists of a fairly simple combination of gradient-based and recurrence-based methods. While this is not necessarily a limitation, it necessitates a thorough set of experiments to justify the combination of approaches. This is the second limitation of the paper: the experimental evaluation is very limited. The authors test of very simple benchmark problems compared to recent work in meta-RL. Moreover, there are few comparisons to baseline methods (especially PEARL) and the authors should include ablation experiments in which they examine the performance of their method relative to strictly gradient-based and strictly recurrence-based methods, using the SAC algorithm as an underlying algorithm.

Finally, the writing of the paper is extremely sloppy. Trials is misspelled as trails throughout the paper. There are also numerous typos, such as "gardient", "meta-reinforce", "Substitue", "apdated", etc. The presentation of the algorithm is quite unclear and the discussion of related work is quite limited.

Overall, the paper should be tested on a wider range of environments and against more competitive baselines to warrant acceptance. The authors should also improve the presentation of the paper to improve clarity.

---

### Decision · Program_Chairs · 2021-01-07
**Final Decision**

**Decision:**

Reject

**Comment:**

A meta-RL algorithms that aims to improve meta policy interpretability by reducing the meta-gradient variance and bias estimation. The method is evaluated on an exploration in 2d navigation and meta-RL benchmarks.

Despite an important topic of research, the reviewers are unanimous that the paper is an early version and requires further work to be suitable for publishing. Specifically, the future versions of the manuscript should address the novelty by better distinguishing from the prior work, improve the evaluations, presentation of the work.